# Effect of Different Monochromatic LEDs on the Environmental Adaptability of *Spathiphyllum floribundum* and *Chrysanthemum morifolium*

**DOI:** 10.3390/plants12162964

**Published:** 2023-08-16

**Authors:** Yinglong Song, Weichao Liu, Zheng Wang, Songlin He, Wenqing Jia, Yuxiao Shen, Yuke Sun, Yufeng Xu, Hongwei Wang, Wenqian Shang

**Affiliations:** 1Zhengzhou Key Laboratory for Research and Development of Regional Plants, College of Landscape Architecture and Art, Henan Agricultural University, Zhengzhou 450002, China; edward_song1989@163.com (Y.S.); liuweichao1991@126.com (W.L.); wzhengt@163.com (Z.W.); yxshen@henau.edu.cn (Y.S.); yukesun1@163.com (Y.S.); xuyufeng_2011@163.com (Y.X.); whw155144@163.com (H.W.); 2School of Horticulture Landscape Architecture, Henan Institute of Science and Technology, Xinxiang 453003, China; jiawq2022@hist.edu.cn

**Keywords:** UV-A, blue light, green light, red light, antioxidant enzyme, photosynthesis

## Abstract

Light-emitting diodes (LEDs) can be programmed to provide specialized light sources and spectra for plant growth. UV-A (397.6 nm), blue (460.6 nm), green (520.7 nm), and red (661.9 nm) LED light sources were used to study the effects of different monochromatic lights on the growth, antioxidant system, and photosynthetic characteristics of *Spathiphyllum floribundum* ‘Tian Jiao’ (a shade-loving species) and *Chrysanthemum morifolium* ‘Huang Xiu Qiu’ (a sun-loving species). This research revealed that green and blue light could enhance the morphological indicators, Chl a/b, photosynthetic electron transfer chain performance, and photosystem activity of *S. floribundum*, blue and red light could enhance the solution protein, Chl a, and photosynthetic electron transfer chain performance of *C. morifolium*, red and UV-A light viewed the highest SOD and CAT activities of *S. floribundum* (275.56 U·min·g^−1^; 148.33 U·min·g^−1^) and *C. morifolium* (587.03 U·min·g^−1^; 98.33 U·min·g^−1^), respectively. Blue and green light were more suitable for the growth and development of the shade-loving plant *S. floribundum*, while red and blue light were more suitable for the sun-loving plant *C. morifolium*. UV-A light could be used for their stress research. The research revealed the different adaptation mechanism of different plants to light environmental conditions.

## 1. Introduction

Light is one of the most important environmental factors in plant growth and development [1,2]. As an important signal and energy source, light regulates plant photomorphogenesis and photosynthesis [3,4,5]. The fluorescent, metal halide, high-pressure sodium, or incandescent lamps often used to plant growth in commercial production, are costly, inefficient, and cannot be set to a single spectrum of light quality, so they produce light wavelengths that interfere with plant growth and differentiation [6]. The advancement and wider availability of next-generation energy-efficient, solid-state lighting offers the opportunity to use this technology in large-scale applications. Light-emitting diode (LED) technology has proven to be ideal in plant lighting according to its small mass/volume ratio, low energy consumption, long life, and various monochromatic spectrum [7,8]. Concomitantly, there is increasing research on plant growth and development both in monochromatic and polychromatic light environments by LED technologies [7,8,9,10].

Plants show different physiological responses under different LED light conditions [1,7]. The absorption peaks of a plant’s photosynthetic pigments are in the blue and red spectra, both of which significantly influence plant growth and development [11,12]. Blue light influences chlorophyll synthesis, leaf anatomical structure, and antioxidant enzyme activities in plants [13,14]. Red light not only drives photosynthesis but also induces stem elongation, soluble sugar synthesis, and changes in photosynthetic apparatus [1,8,15]. The underlying mechanisms of green and ultraviolet (UV) wavelengths might still require further investigation; however, they have been studied significantly in recent years [16,17]. Two bands of UV light, UV-B (280–320 nm) and UV-C (100–280 nm), induce the production of free radicals, which can destroy plant DNA, proteins, lipids, chloroplasts, and photosynthetic pigments [18,19] and can inhibit plant photosynthesis and growth [18,20,21]. Other studies have found that near-UV light (300–400 nm) or UV-A (320–400 nm) can promote plant growth and physiological activity [22,23]. Some researchers pointed out that the chlorophyll in plant leaves absorbs very little green light, which, compared with blue and red light, has little effect on plant photosynthesis [24,25,26]. However, other studies have shown that green light affects plant morphogenesis and physiological responses, including leaf development, stomatal conductance, and early stem elongation [25,27]. Therefore, the monochromatic light at discreet wavelengths may promote or inhibit plant physiological responses by different methods, and the mechanisms behind such light responses require further study.

Many researchers have shown that light quality can affect plant photosynthesis, photosynthate accumulation, and morphogenesis through regulation of antioxidant enzyme activities related to plant growth and development, photosystem II (PSII) electron transport, and other physiological and biochemical processes [28,29,30]. The relationship between PSII and redox balance is a complex physiological and metabolic process. PSII is a pigment protein complex composed of more than 25 subunits on the thylakoid membrane [31]. PSII is very sensitive to changes in light quality. Absorption of the photons in the visible light spectrum by the PSII antenna leads to excitation of an electron in the Photosystem II primary donor P680. P680 is formed as this excited electron is passed through a series of intermediates, Q_A_ and Q_B_, PQ library, Cytb6t and PC, and finally to the PSI receptor, after which carbon fixation reactions occur. P680^+^ is then reduced by an electron from water, which results in the generation of oxygen. However, electrons released by PSII often leak during the transfer to PSI, and these electrons are transferred to molecular oxygen, generating reactive oxygen species (ROS). Accumulated ROS can participate in the destruction of the D1 protein of PSII, inactivating or even irreversibly destroying the PSII reaction center, thereby affecting PSII electron transfer, and ultimately causing photosynthesis to fail. However, plants do have defense mechanisms to balance ROS, such as water–water cycle, cyclic electron flow, and LHCII relocation.

Antioxidant enzymes such as peroxidase (POD), superoxide dismutase (SOD), catalase (CAT), and ascorbate peroxidase (APX) are important protective systems within plants that remove ROS such as the superoxide anion (O^2−^), hydrogen peroxide (H_2_O_2_), and hydroxide ions (HO^−^) [32]. The antioxidant enzyme system in a plant can usually maintain the dynamic balance of free radical production and scavenging in vivo, ensuring the electron transfer of PSII and allowing photosynthesis to proceed normally. Studies have shown that LED light activates the antioxidant enzyme system in plants, initiates synthesis of antioxidant enzymes, and reduces the excessive ROS produced by photosynthesis on thylakoid membranes [13].

That having been said, plants differentially adapt to the environment, which include different light qualities and plant variety types. It is not clear whether monochromatic light will affect the PSII electron transfer and the antioxidant system dynamic balance, it is also unclear how the growth of a variety of types of plant will respond to different monochromatic light treatments. Plants were divided into shade-loving plants and sun-loving plants based on their sensitivity to light [33,34]. Therefore, the shade-loving plant *S. floribundum* and the sun-loving plant *C. morifolium* were chosen as the plant material in the research, which have good ornamental value, high scientific research value, and great market application prospects [35,36]. The plants were exposed to four qualities of monochromatic LED light, namely UV-A light, blue light, green light, and red light, to study the growth, chlorophyll, antioxidant enzyme activities, photosynthesis, and PSII activity to identify the growth response strategies of the two plants under monochromatic light. This report explores the growth response strategies of shade-loving and sun-loving plants to monochromatic LED light sources, which can provide a reference for the analysis of plant growth regulation mechanisms and the application of plant factory production.

## 2. Materials and Methods

### 2.1. Plant Material

Plant materials were obtained from commercial growers. *S. floribundum* ‘Tian Jiao’ (tissue-culture propagated plantlets 3–4 cm in height with 5–6 leaves) were obtained from Guangzhou Fun Gardening Co., Ltd., (Guangzhou, China). Rooted cuttings (5–6 cm in height with 6–7 leaves) of *C. morifolium* ‘Huang Xiu Qiu’ were obtained from Jiangsu Suqian Huamu Co., Ltd., (Suqian, China). The roots were washed with distilled water before the plants were re-planted in a perlite: vermiculite mix (1:1) in plastic pots (12 cm in diameter and 10 cm in depth). There were 20 pots (each pot with one plant) in each treatment and each treatment was repeated three times.

The plants were placed in a light incubator for one week (the environmental conditions in the initial light incubator were maintained with a 12 h photoperiod, a photosynthetic photon flux density of 50 µmol·m^−2^·s^−1^, and a temperature of 23 ± 2 °C), and then transferred to 1 of 4 different light environments (each photosynthetic photon flux density was set at 50 µmol·m^−2^·s^−1^) provided by programmed LEDs (Zhongshan Liangfeng Lighting Appliance Co., Ltd., Zhongshan, China). The LED light sources were programmed for ultraviolet (UV-A, peak at 397.6 nm), blue (B, peak at 460.6 nm), green (G, peak at 520.7 nm), and red (R, peak at 661.9 nm). Each monochromatic light is a control to each other. The LED spectrum analysis was determined by a spectroradiometer (OHSP350P, Hangzhou Hopoo Light & Color Technology Co., Ltd., Hangzhou, China). Figure 1 shows the relative spectral photon for each lighting source, UV-A and blue LEDs had a small overlap in the spectrum range of 417.5 nm–440 nm, blue and green LEDs had a small overlap in the spectrum range of 462.5 nm–530 nm. The growth chambers were maintained with a 12 h photoperiod, a photosynthetic photon flux density of 50 µmol·m^−2^·s^−1^, and a temperature of 23 ± 2 °C. Plants were irrigated with 1/3 Hoagland [37] nutrient solution every three days.

### 2.2. Plant Growth

After 60 days of growth, 15 plants were selected from each treatment for determination of fresh weight, dry weight, plant height, leaf number, and maximum root length. The plant height and root length were measured by the vernier scale. After the fresh weight was determined, the material was dried in an oven at 105 °C for 30 min, then at 60 °C for 48 h, and then the dry weight was determined [38], both using a 1/10,000 balance scale (FA2104B, Shanghai Precision Scientific Instrument Co., Ltd., Shanghai, China). The fresh weight/dry weight ratio was calculated. 

### 2.3. Photosynthetic Pigments

Photosynthetic pigments from leaf samples were determined according to the method of Holm [39], 0.2 g of chopped leaves were placed in 20 mL of a 1:1 (*v*/*v*) mixture of 80% acetone (acetone:water = 80:20, *v*/*v*) and absolute ethyl alcohol in a 25 mL stoppered test tube in the dark for 24 h. Using 80% acetone as the blank, the absorbance (OD) was measured at λ = 663, 645 and 470 nm using a UV spectrophotometer (UV-1900, Shimadzu, Tokyo, Japan). The chlorophyll a/b value was calculated. 

### 2.4. Soluble Sugar and Soluble Protein

Soluble sugars were extracted using the method of Fairbairn [40] with slight modifications. Leaves (0.2 g) were put into a test tube, to which 10 mL of distilled water was added and mixed. After 30 min in a water bath at 100 °C, the supernatant was collected, and distilled water was added to a volume of 25 mL. The soluble sugar content was determined with the sulfuric acid anthrone method at a wavelength of 620 nm using a UV spectrophotometer (UV-1900, Shimadzu, Tokyo, Japan).

Leaves (0.2 g) were placed in a mortar, flash frozen with liquid nitrogen, and ground to a powder. Phosphate buffer (5 mL; 50 mM, pH 7.0) was added, and the homogenate was centrifuged at 12,000 rpm for 4 min at 4 °C. The supernatant was used for determination of soluble protein content according to the method of Bradford [41], using 0.1 mL of the supernatant and 4.9 mL of Coomassie Brilliant Blue G-250 (0.1 g·L^−1^). After 2 min, the absorbance was measured at 595 nm using a UV spectrophotometer (UV-1900, Shimadzu, Tokyo, Japan), and the soluble protein content was calculated using a standard curve (bovine serum albumin was used to make the standard curve). 

### 2.5. Antioxidant Enzyme Activities

Activities of four antioxidant enzymes were determined by spectrophotometric methods. Leaf samples (0.2 g) were ground in a mortar with liquid nitrogen, to which 5 mL of phosphate buffer (50 mM, pH 7.8) containing 1% polyvinyl pyrrolidone (PVP), 0.2 mM ethylene diamine tetraacetic acid (EDTA), and 5 mM ascorbic acid (ASA) was added. The slurry was centrifuged at 12,000 rpm at 4 °C for 20 min. The supernatant was used for determination of the enzyme activities.

Superoxide dismutase (SOD) activity was measured by photochemical reduction of nitrotetrazolium blue chloride (NBT) [42]. The enzyme extract from above (0.05 mL) was mixed with 1.5 mL phosphoric acid buffer (50 mM, pH 7.8), 0.3 mL Met solution (130 mM), 0.3 mL NBT solution (0.75 mM), 0.3 mL EDTA solution (0.1 mM), 0.3 mL riboflavin solution (0.02 mM), and 0.25 mL distilled water in the reaction tube. The control reaction contained enzyme extract and only 0.05 mL phosphoric acid buffer. After mixing, one reaction tube was placed in the dark as a positive control, and the rest of the reaction tubes were centrifuged at 12,000 rpm for 20 min. At the end of the reaction, the absorbance value was measured at 560 nm wavelength, and 1 unit (U) of SOD activity was the amount of enzyme that inhibited the photochemical reduction of NBT by 50%. 

The analysis of peroxidase (POD) activity was based on the oxidation of guaiacol using H_2_O_2_ according to the method described by Zhang and Kirham [43]. The reaction mixture consisted of buffered pyrogallol in 0.1 M potassium phosphate buffer (pH 7.0), 1% H_2_O_2_, and the enzyme extract. The reaction rate was calculated by monitoring the changes in absorbance at 430 nm (ε = 12 mM^−1^·cm^−1^), with the activity expressed as U·mg^−1^ (protein)·min^−1^.

Catalase (CAT) activity was measured according to the method described by Cakmak and Marschner [44] by reading the absorbance value of the reaction solution containing phosphate buffer (25 mM, pH 7.8), H_2_O_2_ (10 mM), and enzyme solution (0.2 mL) at 240 nm. As such, 1 U CAT activity was the amount of enzyme that reduced the OD_240_ value by 0.01 per min. 

Ascorbate peroxidase (APX) activity was determined with reference to the Nakano and Asada [45] ascorbate (Asc) method. The reaction mixture consisted of 50 mM phosphoric acid buffer (pH 7.0), 5 mM Asc, 20 mM H_2_O_2_, and 5 mM EDTA. After the enzyme extract was added to the above mixture, the absorption values were immediately measured at 240 nm over 2 min at 20 s intervals.

### 2.6. Photosynthetic Parameters

Net photosynthetic rate (A), transpiration rate (Tr), stomatal conductance (gs), intercellular CO_2_ concentration (Ci), vapor pressure deficit (VPD), and water use efficiency (WUE) were measured by a photosynthetic apparatus (CIRAS-3, Hansatech Instruments Ltd., Norfolk, UK). The leaf temperature was 24.7 ± 0.8 °C and the ambient CO_2_ concentration was 450 ± 5 µmol·mol^−1^. The measurements were measured on 5 randomly selected plants under each light condition for each species.

### 2.7. Chlorophyll Fluorescence

The fast chlorophyll fluorescence induction kinetic curves (OJIP curve) of *C. morifolium* and *S. floribundum* were assayed by a multifunctional plant efficiency instrument (m-pea-1, Hansatech Instruments Ltd., Norfolk, UK), the plants should be dark-adapted for 30 min before they are measured. The OJIP curve was induced by a 3000 mol·m^−2^·s^−1^ pulse of light, and the fluorescence signal was detected from 10 positions for 1 s, with an initial rate of 105 data points per second. Then, we analyzed the OJIP curve using the jip-test [46]. The following fluorescence parameters were obtained: the light energy absorbed by the unit reaction center (ABS/RC); the energy (TR_o_/RC) captured by the unit reaction center for the reduction of QA; the energy captured by the unit reaction center for electron transport (ET_o_/RC); the energy dissipated by unit reaction center (DI_o_/RC); the probability that the captured exciton would transfer electrons to other electron receptors in the electron transport chain that exceed QA (ψ_o_); the quantum yield of the terminal electron acceptor phi Ro (φ_Ro_); the quantum ratio for heat dissipation (φ_Do_); the quantum yield for electron transfer (φ_Eo_); the number of reactive reaction centers per unit area (RC/CS_o_); the performance index based on absorption of light energy (PI_abs_); and the comprehensive performance index (PI_total_). The measurements were repeated 5 times per species for each light condition.

### 2.8. Statistical Analysis

Data underwent a one-way analysis of variance (ANOVA), and significant differences between the means were tested using Duncan’s post-hoc test (*p* ≤ 0.05) with the SPSS 19.0 software (IBM, Inc., Chicago, IL, USA).

## 3. Results

### 3.1. Growth Characteristics

After 60 days of cultivation under different monochromatic light spectra, LED light quality did result in significant differences in total fresh weight, total dry weight, plant height, leaf number, and maximum root length in both *S. floribundum* and *C. morifolium* (Table 1, Figure 2). The total fresh weight values for both species were significantly greater under blue LED than under green (by 81.80% for *S. floribundum* and 27.20% for *C. morifolium*), red (97.11% and 39.16%), or UV-A (272.64% and 309.84%). The total dry weight values of *S. floribundum* and *C. morifolium* were also significantly greater under blue LED than under either green (by 95.56% and 32.37%), red (by 57.14% and 33.33%), or under UV-A (by 282.61% and 360.00%). The results showed that blue light was more favorable to *S. floribundum* and *C. morifolium* biomass accumulation than other monochromatic light, while UV-A light was not favorable to biomass accumulation. The height of *S. floribundum* was greatest under blue LED, followed by green LED, which both yielded plants significantly taller than UV-A or red LEDs. The maximum height value for *C. morifolium* also occurred under blue LED, followed by the green and red spectra. The results showed that blue light was more favorable to stem elongation in both *S. floribundum* and *C. morifolium* than other monochromatic light. The number of *C. morifolium* leaves was greatest in both red and blue LED. The results showed that blue light significantly promoted the number of leaves in *S. floribundum*, while red light and blue light increased the number of leaves in *C. morifolium*. UV-A light resulted in the lowest number of leaves in *S. floribundum* and *C. morifolium* among these monochromatic light treatments. The maximum root length values for both *S. floribundum* and *C. morifolium* occurred when the plants were grown under the green LED, followed by the blue and red LEDs, with the minimum values under the UV-A LED.

### 3.2. Soluble Sugars, Soluble Proteins, and Chlorophyll

Growth under different wavelengths of monochromatic light affected the synthesis and accumulation of soluble sugar and protein in *S. floribundum* and *C. morifolium* plants (Table 2). The soluble sugar content in *S. floribundum* was significantly different under each monochromatic light, with significantly higher content in plants grown under red or blue LED than under green (by 140.24% and 84.62%) or UV-A (by 301.98% and 208.91%). The soluble sugar content in *C. morifolium* under different light showed the opposite effect, that is, the soluble sugar content under green or UV-A LED were higher than those under blue or red LED. It indicated that the synthesis and accumulation of soluble sugar in the two plants are different in response to monochromatic radiation. The highest soluble protein level in *S. floribundum* was under red LED, followed by blue LED. The soluble protein level in *C. morifolium* was the highest under blue LED, followed by red LED, which were significantly higher than the levels under green LED. The results showed that red and blue light better promoted the synthesis of soluble proteins in both *S. floribundum* and *C. morifolium* than green or UV-A light. 

The content of chlorophyll a and chlorophyll b in *S. floribundum* was higher under red or green LED than under blue or UV-A spectra. Red and green light promoted the synthesis of chlorophyll a and b in *S. floribundum*, while blue light promoted the synthesis of chlorophyll a and b in *C. morifolium*. UV-A light inhibited the synthesis of both chlorophyll pigments in both *S. floribundum* and *C. morifolium* compared with the other monochromatic lights. In *S. floribundum,* there were no significant differences in the content of carotenoids or the chlorophyll a/b ratio under the different light treatments, while red light did yield a significantly higher carotenoid content and chlorophyll a/b ratio in *C. morifolium*.

### 3.3. Activity Levels of Antioxidant Enzyme

In both *S. floribundum* and *C. morifolium*, SOD, CAT, POD, and APX showed varying degrees of activities under the different bands of monochromatic light (Figure 3). In *S. floribundum*, every light condition except blue LED induced the highest activity of at least one enzyme. SOD activity was highest under red LED and lowest under blue LED. CAT activity was highest under red LED and lowest under green LED. POD activity was significantly higher under green LED than under any of the other monochromatic light treatments, while APX activity was highest under UV-A treatment, and lowest in red treatment. In *C. morifolium*, SOD and CAT activities were highest under UV-A LED, and lowest under red or blue LED. POD activity was highest under red LED, and lowest with UV-A treatment. Green LED yielded the highest activity APX, while blue LED the lowest. In *C. morifolium* leaves, UV-A LED best enhanced SOD and CAT activity, red LED best promoted POD activity, and green LED most increased APX activity.

### 3.4. Photosynthetic Characteristics

In both *S. floribundum* and *C. morifolium*, monochromatic light had differential effects on the net photosynthetic rate (Pn), intercellular CO_2_ concentration (Ci), transpiration rate (Tr), stomatal conductance (Gs), vapor pressure deficit (VPD), and water use efficiency (WUE) (Figure 4). The lack of significant effect of different LED spectra on vapor pressure deficit was the only similar response between the two species (Figure 4E). These results showed that the monochromatic light had no significant effect on the vapor pressure deficit of *S. floribundum* and *C. morifolium*. All other measured photosynthetic parameters showed different effects under the different light spectra between the two species.

In *S. floribundum*, the Pn was significantly higher under blue light than under the other monochromatic light treatments (Figure 4A). But the Ci was higher under UV-A and red light (Figure 4B). The Tr and Gs were highest under blue light (Figure 4C,D), similar to that of the Pn. Together, these parameters indicate that blue light has a significant effect on photosynthesis in the leaves of *S. floribundum*, and that as the Pn increases, so does the Tr through opening the stomata, which leads to a decrease in Ci. The maximum WUE in *S. floribundum* was significantly higher under blue and green light, as compared to UV-A light, but was still the highest under blue light treatment. While blue and green light had a stronger effect on WUE, blue light was still more effective at promoting photosynthesis, possibly because shade plants like *S. floribundum* prefer blue light.

In *C. morifolium* leaves, the Pn was significantly higher under green light than that of blue, and green and blue light significantly promoted photosynthesis, while red light increased it compared to UV-A light (Figure 4A). Again, the Ci response under the different light qualities was opposite to the Pn (Figure 4B). The highest Tr and maximum Gs appeared under red light treatment, followed by green treatment, both of which were significantly higher than blue and UV-A treatment. Again, the WUE in *C. morifolium* was similar to the Pn, with the largest value under green light treatment, followed by blue light treatment. In the sun-loving *C. morifolium*, green light had a stronger effect on Pn and WUE, red and green light could promote Tr and Gs, and red light increased the Ci, again an opposite response compared to the Pn.

### 3.5. Fast Chlorophyll Fluorescence

Different monochromatic light qualities had different effects on the fast chlorophyll fluorescence parameters of *S. floribundum* and *C. morifolium* (Table 3, Figure 5). Under red LED, the electron transfer probability (ψ_o_) of the Q_A_-downstream and the quantum yield for electron transfer (φ_Eo_) in the electron transfer chain of *S. floribundum* leaves was greatly reduced, and that was used to reduce the quantum yield of terminal electron acceptor at the PS I receptor side (φ_Ro_) and the quantum ratio (φ_Do_) used for heat dissipation was significantly increased, eventually resulting in a decrease in maximum photochemical efficiency (φ_Po_). This indicated that, in *S. floribundum* under red light, electron transfer is severely inhibited, the quantum yield and photochemical efficiency are significantly decreased, and the relative activity of PS I is significantly affected.

The rapid chlorophyll fluorescence parameters of *S. floribundum* leaves under green and blue lights were exactly the opposite of those under red light treatment, that is, ψ_o_, φ_Eo_, and φ_Ro_ at the PS I receptor side were significantly increased, while φ_Do_ was significantly reduced. Finally, φ_Po_ was increased. The results showed that the electron transfer of *S. floribundum* under green and blue light were severely promoted, and the quantum yield and photochemical efficiency were significantly increased. 

In *C. morifolium*, ψ_o_ and φ_Eo_ was significantly lower under UV-A treatment than under the other monochromatic light bands. φ_Ro_ was the lowest, φ_Do_ was the highest. Under blue light treatment, φ_Eo_, ψ_o_, and φ_Po_ were the highest, while φ_Do_ was lowest.

In *S. floribundum*, red light was significantly higher than the light energy absorbed by the unit reaction center (ABS/RC), the light energy dissipated per unit reaction center (DI_o_/RC), and the light energy captured by the unit reaction center (TR_o_/RC) was compared to that under blue and green light (Table 4). However, under blue and green light, the light energy used for electron transfer in the unit reaction center (ET_o_/RC) and the light energy delivered to the PS I in the unit reaction center (RE_o_/RC) of *S. floribundum* were not significantly different with that under red light. Under blue and green light, the performance index (PI_abs_) and the comprehensive performance index, based on the absorption of light energy (PI_total_) of *S. floribundum* were significantly higher than under UV-A or red light treatment. 

Under blue light, ABS/RC, DI_o_/RC, TR_o_/RC, and ET_o_/RC of *C. morifolium* was significantly lower than green light, while PI_abs_ and PI_total_ of *C. morifolium* were significantly higher under blue light than under green light. Under red light, ABS/RC, DI_o_/RC, TR_o_/RC, and ET_o_/RC of *C. morifolium* were not significantly different with under UV-A and green light, while PI_abs_ and PI_total_ of *C. morifolium* were significantly higher under red light than under UV-A and green light.

### 3.6. Correlation Analysis

The correlation analysis of physiological indicators in *S. floribundum* was shown in Figure 6A. The morphological indicators had a significant positive correlation with Chl a/b, Pn, Tr, Gs, WUE, ψ_o_, φ_Eo_, φ_Ro_, ET_o_/RC, RE_o_/RC, PI_abs_, and PI_total_, but they showed a significant negative correlation with SOD and Ci. Soluble sugar and soluble protein had a significant positive correlation with Chl a, Chl b, CAT, φ_Do_, ABS/RC, and DI_o_/RC, and they were significantly negatively correlated with carotenoid, APX, and φ_Po_. Chl a and Chl b showed a significant positive correlation with VPD, φ_Do_, ABS/RC, and DI_o_/RC, and they were significantly negatively correlated with carotenoid, APX, and φ_Po_. The carotenoid was significantly positively correlated with APX and Ci. The SOD activity was significantly positively correlated with Ci, VPD, ABS/RC, DI_o_/RC, and TR_o_/RC, and significantly negatively correlated with Chl a/b, Pn, Tr, Gs, WUE, ψ_o_, φ_Eo_, φ_Ro_, ET_o_/RC, RE_o_/RC, PI_abs_, and PI_total_. The CAT showed a similar correlation trend with SOD, but the POD overall showed an opposite trend. The Pn, Tr, and Gs were significantly positively correlated with Chl a/b, WUE, and RE_o_/RC, and they significantly negatively correlated with SOD and Ci.

The correlation analysis of physiological indicators in *C. morifolium* was shown in Figure 6B. The morphological indicators (excluding root length) were significantly positively correlated with soluble protein, Chl a, Chl b, POD, Pn, ψ_o_, φ_Eo_, φ_Ro_, RE_o_/RC, PI_abs_ and PI_total_, but they showed a significant negative correlation with SOD, CAT, Ci, φ_Do_, ABS/RC, DI_o_/RC, and TR_o_/RC. Soluble sugar had a significant positive correlation with WUE, φ_Do_, ABS/RC, DI_o_/RC, TR_o_/RC, and ET_o_/RC, and significantly negatively correlated with Ci, φ_Po_, φ_Ro_, PI_abs_ and PI_total_. The correlation between soluble protein and various indicators overall showed a opposite trend with soluble sugars. Chl a and Chl b were significantly positively correlated with soluble protein, POD, Pn, φ_Po_, ψ_o_, φ_Eo_, φ_Ro_, RE_o_/RC, PI_abs_ and PI_total_, but they showed a significant negative correlation with SOD, CAT, φ_Do_, ABS/RC, DI_o_/RC, and TR_o_/RC. The carotenoid showed a significant positive correlation with POD, Tr, Gs and RE_o_/RC. Chl a/b was significantly positively correlated with SOD and VPD. SOD was only significantly positively correlated with VPD. The correlation between POD and various indicators overall showed a opposite trend with SOD. CAT and APX viewed a significant positive correlation with φ_Do_, ABS/RC, DI_o_/RC, TR_o_/RC, and ET_o_/RC. Pn was significantly positively correlated with WUE, ψ_o_, φ_Eo_, and RE_o_/RC, and negatively correlated with Ci. Tr and Ci had a significant positive correlation with carotenoid, Chl a/b, POD, APX, and RE_o_/RC, but viewed a significant negative correlation with SOD and VPD.

## 4. Discussion

Recent studies have used the LED light technologies to test how plants respond to monochromatic light, because low costs and wide availability of these technologies open up the possibility of culturing an entire greenhouse under augmented light spectra [29,47]. For instance, in cherry tomato seedlings, the fresh and dry weight of plant buds and roots, the leaf area, and the stem diameter were greater under single-spectrum green light than under other monochromatic light treatments [48]. Alvarenga et al. [27] studied the growth and development of tissue culture plantlets of *Achillea millefolium* by red light, blue light, white light, green light, and far-red light. The fresh and dry weights, plant height, and leaf number were greater for plants grown under blue light compared with the other monochromatic spectra. In this experiment, the fresh weight, dry weight, and plant height of *S. floribundum* and *C. morifolium* were the greatest under blue light. These qualities were significantly higher in plants grown under blue light than under the other monochromatic light treatments, with UV-A treatment producing the smallest plants by all measures. Seeing the best growth under blue light was similar to results for lettuce [49], *Achillea millefolium* [27], and *Rehmannia glutinosa* [50]. This phenomenon was similar to related research in *Nicotiana tabacum* and *Pisum sativum* [51,52]. It may be due to differences in the effects of different wavelengths of light on the functional status of photosynthetic organs and pigment protein synthesis, while blue light can participate in and actively coordinate the functional relationship between the nucleus and plastids. The growth quality of *S. floribundum* and *C. morifolium* under green light is only inferior to blue light, indicating that green light plays an important role in promoting the morphogenesis of the two plants. It may be that green light is also involved in the synergistic regulation of chlorophyll, which resulted in a higher photosynthetic efficiency.

Light quality directly affects the synthesis of photosynthetic pigments, which affect plant photosynthesis, synthesis, and the accumulation of metabolites [53,54]. Some studies have shown that blue light affects the content of chlorophyll in leaves [12,55]. The contents of chlorophyll a and chlorophyll b in *C. morifolium* leaves in this experiment were higher under blue light, which is consistent with the result on hemerocallis [56]. The contents of chlorophyll a and chlorophyll b in *S. floribundum* leaves were lower under blue light than that of red or green light. But the net photosynthetic rate was higher under blue light. This phenomenon was similar to the research on *Lactuca sativa* [57], it may be due to the fact that plants with less chlorophyll content are more effective at using chlorophyll than plants with excess chlorophyll. *S. floribundum* showed more chlorophyll under red and green light, possibly to compensate for the decrease in photosynthetic rate caused by insufficient chlorophyll activity. The content of soluble protein and soluble sugar in *S. floribundum* is the highest under red light. *C. morifolium* achieve their maximum values under blue and green light, respectively. Both the changing trends chlorophyll, soluble protein, and soluble sugar contents in *C. morifolium* and *S. floribundum* all perfectly interpret the dynamic regulation mechanism of photoresponse systems under different monochromatic light conditions.

Plants have complex and dynamic photoresponse systems, involving reactive oxygen and hormonal signals, for optimizing light adaptation and defense systems [58]. As a stimulating factor, light activates antioxidant defense systems in plants and synthesizes antioxidants [13]. As the first line of defense in antioxidant enzyme systems, SOD specifically converts superoxide anion radicals into hydrogen peroxide and oxygen molecules, and several studies have shown that elevated SOD activity is associated with plant tolerance to environmental stress [59,60]. In *C. morifolium* grown under UV-A light, SOD activity was higher, creating a higher active oxygen scavenging ability. It may be that UV-A light stimulates plants to produce excess superoxide anion, which has been recognized as a signaling factor induced by antioxidant enzymes that stimulates the production of SOD. The UV-A light also induced high CAT activity in *C. morifolium*. This high CAT activity is likely in response to the higher SOD activity under UV-A light, because the conversion of superoxide anion radicals by SOD produces a large amount of hydrogen peroxide, which is removed by CAT. But *S. floribundum* viewed the maximum value of SOD and CAT under red light. As a highly active, adaptive enzyme, POD can reflect the characteristics of plant growth and development, the metabolic state, and the adaptability to the external environment [61]. The POD activity in *S. floribundum* was stronger under green light, while the POD activity in *C. morifolium* was stronger under red light. This indicates that *S. floribundum* was more adaptable to the environment under green light (such as what filters through the canopy of plants above this shade-tolerant species), while *C. morifolium* was more suitable for growth under red light (which it would be exposed to as a sun-tolerant species). This phenomenon could be interpreted using the research of cowpea [62], which showed that an increase in POD activity may be primarily involved in regulating plant growth, rather than protecting plant tissues from oxidative damage caused by hydrogen peroxide. APX is a key enzyme in the ascorbate–glutathione cycle, using electrons from photosynthetic organs or NADPH as a reducing force to remove H_2_O_2_ from chloroplasts and to modulate the signaling of reactive oxygen intermediates [60]. APX activity was higher under UV-A light in *S. floribundum*, but higher under green light in *C. morifolium*, indicating that *S. floribundum* had a greater ability to scavenge ROS in chloroplasts under UV-A light, while *C. morifolium* had a stronger ability to clear ROS under green light.

Plant photosynthetic regulation mainly includes regulating photosynthetic pigments, chloroplast structure, and stomatal movement [1,63]. It was found that blue light can affect plant photosynthesis by inducing the opening of stomata [54,64,65]. In this experiment, blue light significantly increased the net photosynthetic rate, transpiration rate, and stomatal conductance of the leaves of *S. floribundum*, mainly because blue light induced stomatal opening. The chlorophyll content in *S. floribundum* leaves was the highest under red light, but the net photosynthetic rate was significantly decreased. This may be because the soluble sugar content of *S. floribundum* leaves was higher under red light treatment, and the photosynthetic products were inhibited from being exported from leaves, resulting in a significant decrease in photosynthetic rate through negative feedback. In addition, the stomatal conductance in *S. floribundum* leaves decreased significantly under red light, while the intercellular CO_2_ concentration increased significantly, indicating that the utilization efficiency of carbon dioxide was lower under red light and that the limiting factors for photosynthetic rate decline are non-stomatal factors. This phenomenon was similar with the research by Farquhar [66], and it may be due to a decrease in the photosynthetic activity of mesophyll cells. Most studies have found that plant leaves absorb very little green light, so green light is often considered ineffective for plant photosynthesis and not conducive to plant growth [16,24,25,26]. However, relevant studies have shown that green light can affect the proportion and state of light distribution in plant leaves, participating in the regulation process of photosynthesis [67,68,69]. In this experiment, the net photosynthetic rate of *C. morifolium* under green light was significantly higher than under other monochromatic light, probably because red light and blue light were effectively absorbed by the plant surface, while green light may have reduced the potential light gradient inside the leaf and provided the energy for photosynthesis in deeper layers. Researchers have shown that chloroplast decline would occur because part of the membrane system was destroyed and because stomatal conductance was decreased, which would increase the intercellular carbon dioxide concentration, especially because carbon dioxide fixation was blocked, and a large amount of reactive oxygen species would accumulate [19,22,70]. Compared with other monochromatic light, the net photosynthetic rates of *S. floribundum* and *C. morifolium* were the lowest under UV-A light, which may be caused by UV-A light irradiation destroying the chloroplast system, causing the chlorophyll a, chlorophyll b, and chlorophyll a/b values to decrease, and ultimately the number of chloroplasts to decrease. Although the activity of POD and CAT in *S. floribundum* and *C. morifolium* leaves was higher under UV-A light, it was likely not enough to remove the excess active oxygen species in the plant, resulting in blocked photosynthetic electron transport and decreased net photosynthetic rate, which makes the plants unable to grow normally.

The photoreaction phase of photosynthesis mainly absorbs light energy and converts it into active chemical energy through photosystem II (PSII) and photosystem I (PSI). Photosystem II (PSII) is located on the thylakoid membranes and is the primary site for photosynthesis. Its performance directly determines the light energy utilization efficiency and photosynthesis of leaves [45,71]. The rapid chlorophyll fluorescence kinetics curve (JIP test) is based on the energy flow principle, and can quantitatively explain PSII light energy absorption, conversion, electron transport, PSII action center activity on the receptor side and donor side, and the dynamic changes of the redox state of the electron transporter [45,72]. In the research, it showed that ultraviolet light caused the PSI reaction center (φ_Ro_) in *S. floribundum* and *C. morifolium* leaves to decrease, while the maximum photochemical efficiency (φ_Po_) of the PSII reaction center was not significantly different among the monochromatic light treatments, indicating that the PSI reaction center of *S. floribundum* and *C. morifolium* leaves was more UV-sensitive than the PSII reaction center. In *S. floribundum* leaves under red light, the light energy and the captured light energy absorbed by the unit reaction center and the dissipated light energy increased, while the electron transport capacity of the PSII receptor side decreased and heat loss increased, indicating that red light suppresses PSII electron transfer from the primary receptor (Q_A_ to Q_B_), resulting in a large accumulation of electrons at Q_A_ in *S. floribundum*. However, green and blue light could significantly improve the photosynthetic electron transfer chain performance and net photosynthetic rate of *S. floribundum*; blue and red light carried out the same for *C. morifolium*.

The correlation analysis results also showed that morphological indicators of *S. floribundum* and *C. morifolium* were all significantly positively correlated with Pn, Tr, Gs, ψ_o_, φ_Eo_, φ_Ro_, RE_o_/RC, PI_abs_ and PI_total_. It indicated that the growth strategies of *S. floribundum* and *C. morifolium* under different monochromatic light conditions were closely related to their utilization efficiency of light energy. At the same time, the morphological indicators of *S. floribundum* showed a significant positive correlation with Chl a/b, but the morphological indicators of *C. morifolium* showed a significant positive correlation with Chl a and Chl b. It was consistent with the phenomenon that *S. floribundum* viewed the greatest overground morphological indicators, net photosynthetic rate, transpiration rate, and stomatal conductance of the leaves under blue light, *C. morifolium* showed the highest chlorophyll content and photosynthetic electron transfer chain performance under red light. This phenomenon might be due to the fact that *S. floribundum* was a shade-loving plant, *C. morifolium* was a sun-loving plant, and shade-loving plants could better utilize blue light, while sun-loving plants had a higher utilization rate of red light [73,74]. Meanwhile, there was also a significant positive correlation between indicators such as chlorophyll content, soluble sugar, soluble protein, antioxidant enzyme activity, and photosynthetic fluorescence parameters of *S. floribundum* and *C. morifolium*. This indicated the complexity of light utilization strategies and efficiency of *S. floribundum* and *C. morifolium* under different monochromatic light conditions.

## 5. Conclusions

This research revealed that green and blue light could enhance the morphological indicators, Chl a/b, photosynthetic electron transfer chain performance, and photosystem activity of *S. floribundum*, blue and red light could enhance the solution protein, Chl a, and photosynthetic electron transfer chain performance of *C. morifolium*, red and UV-A light could promote SOD and CAT enzyme activities of *S. floribundum* and *C. morifolium*, respectively. This result indicated that blue and green lights were more suitable for the growth and development of the shade-loving plant *S. floribundum*, while red and blue lights were more suitable for the sun-loving plant *C. morifolium*. UV-A light could be used for their resistance research. Overall, the shade-loving plant *S. floribundum* could fully utilize blue light for photosynthetic metabolism according to its own characteristics, while the sun-loving plant *C. morifolium* could also fully utilize its own advantages to regulate the growth and development process using red light. They could adjust indicators in a timely manner according to light environmental conditions to fully utilize light energy. The research could provide technical and theoretical support for plant factory and plant photo-physiological regulation, however, the specific mechanism was still unclear and needs to be further studied.

## Figures and Tables

**Figure 1 plants-12-02964-f001:**
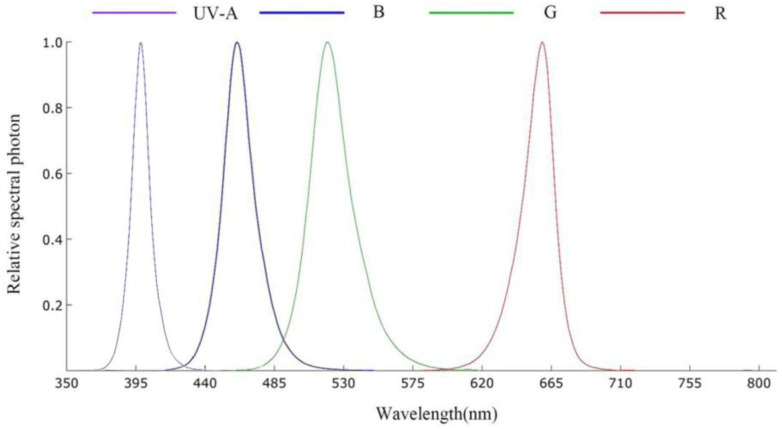
Relative spectral distribution of four light treatments using an LED light source. Ultraviolet (UV-A), blue (B), green (G), and red (R) light. Spectrum was measured at the plant canopy level with an OHSP350P spectroradiometer (Hangzhou Hopoo Light & Color Technology Co., Ltd., Hangzhou, China).

**Figure 2 plants-12-02964-f002:**
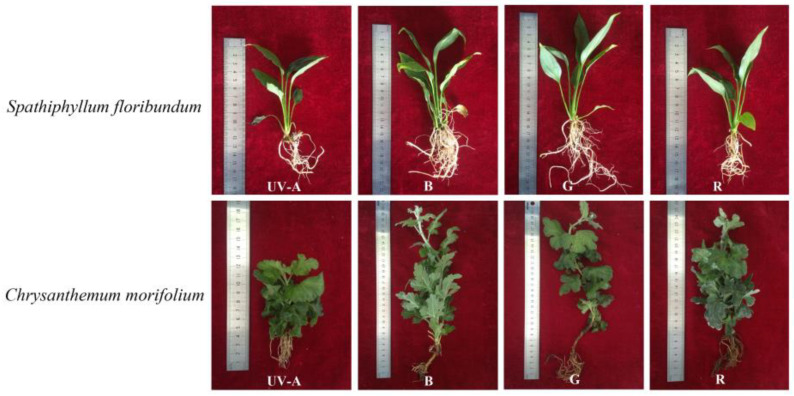
Morphology of *Spathiphyllum floribundum* and *Chrysanthemum morifolium* under different light quality conditions. Ultraviolet (UV-A), blue (B), green (G), and red (R) light.

**Figure 3 plants-12-02964-f003:**
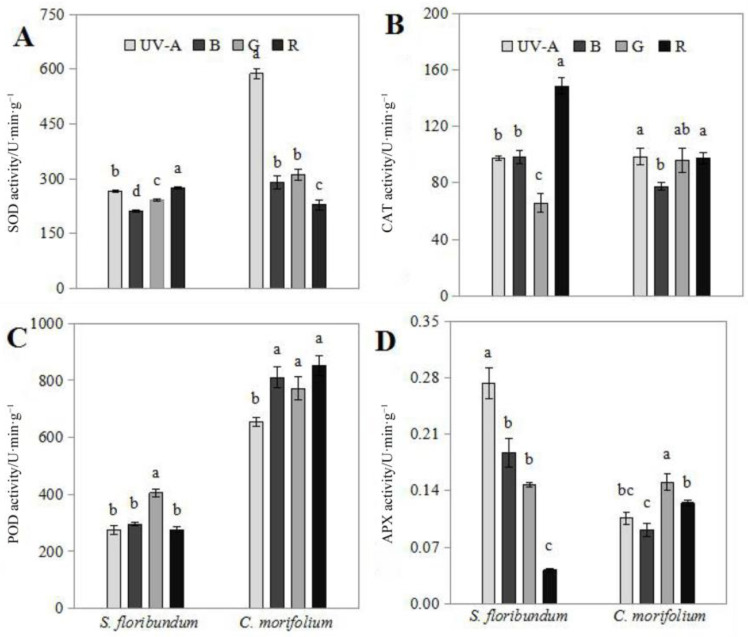
Activities of antioxidant enzymes in *Spathiphyllum floribundum* (*S. floribundum*) and *Chrysanthemum morifolium* (*C. morifolium*) grown under monochromatic LED light. Superoxide dismutase (SOD, (**A**)), peroxidase (POD, (**B**)), catalase (CAT, (**C**)), and ascorbate peroxidase (APX, (**D**)) activities when grown under ultraviolet (UV-A), blue (B), green (G), and red (R) LED light for 60 days. Values are means-standard errors. Means with different letters are statistically different (*p* < 0.05; *n* = 3) as determined by Duncan’s post-hoc test.

**Figure 4 plants-12-02964-f004:**
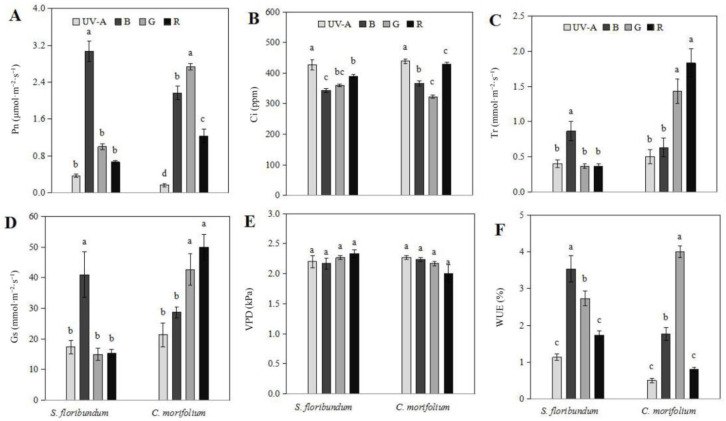
Photosynthetic parameters in *Spathiphyllum floribundum* (*S. floribundum*) and *Chrysanthemum morifolium* (*C. morifolium*) grown under monochromatic light provided by programmable LED. The net photosynthetic rate (Pn, (**A**)), intercellular CO_2_ concentration (Ci, (**B**)), transpiration rate (Tr, (**C**)), stomatal conductance (Gs, (**D**)), vapor pressure deficit (VPD, (**E**)), and water use efficiency (WUE, (**F**)) values in *S. floribundum* (**left**) and *C. morifolium* (**right**) grown under ultraviolet (UV-A), blue (B), green (G), and red (R) LED light for 60 days. Values are means-standard errors. Means with different letters are statistically different (*p* < 0.05; *n* = 3) as determined by Duncan’s post-hoc test.

**Figure 5 plants-12-02964-f005:**
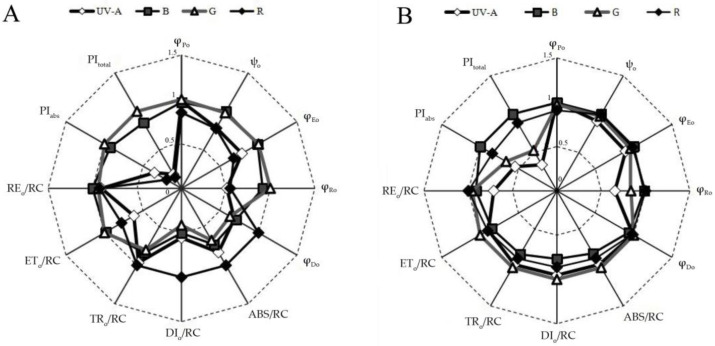
Comparison of the fluorescence parameters as determined for *S. floribundum* (**A**) and *C. morifolium* (**B**) grown under ultraviolet (UV-A), blue (B), green (G), and red (R) LEDs for 60 days. Values are means-standard errors. All the values were normalized (divided by the maximal) to allow comparison of the variables measured on different scales.

**Figure 6 plants-12-02964-f006:**
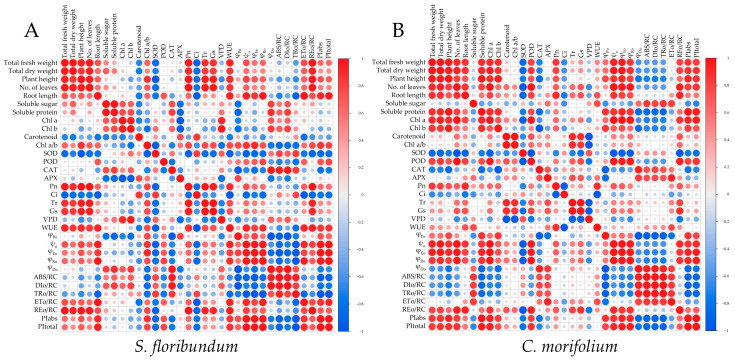
Correlation analysis between physiological parameters of *S. floribundum* (**A**) and *C. morifolium* (**B**). Chlorophyll (Chl), superoxide dismutase (SOD), peroxidase (POD), catalase (CAT), ascorbate peroxidase (APX), net photosynthetic rate (Pn), intercellular CO_2_ concentration (Ci), transpiration rate (Tr), stomatal conductance (Gs), vapor pressure deficit (VPD), water use efficiency (WUE), maximum photochemical efficiency (φ_Po_), electron transfer probability (ψ_o_), quantum yield for electron transfer (φ_Eo_), quantum yield of terminal electron acceptor at the PS I receptor side (φ_Ro_), quantum ratio (φ_Do_), light energy absorbed by the unit reaction center (ABS/RC), light energy dissipated per unit reaction center (DI_o_/RC), light energy captured by the unit reaction center (TR_o_/RC), light energy used for electron transfer in the unit reaction center (ET_o_/RC), light energy delivered to the PS I in the unit reaction center (RE_o_/RC), performance index (PI_abs_), and absorption of light energy (PI_total_). All the values were normalized (divided by the maximal) to allow comparison of the variables measured on different scales.

**Table 1 plants-12-02964-t001:** Growth parameters of *S. floribundum* and *C. morifolium* grown under UV-A, blue, green, and red LEDs for 60 days.

Species	Light Quality	Total Fresh Weight (g)	Total Dry Weight (g)	Plant Height (cm)	No. of Leaves (Per Plant)	Root Length (cm)
*S. floribundum*	UV-A	2.01 d ± 0.04	0.23 d ± 0.01	12.62 c ± 0.18	5.40 c ± 0.24	9.34 c ± 0.20
	B	7.49 a ± 0.09	0.88 a ± 0.03	17.02 a ± 0.22	9.80 a ± 0.37	13.80 a ± 0.26
	G	4.12 b ± 0.06	0.45 c ± 0.02	13.80 b ± 0.23	6.20 bc ± 0.37	14.30 a ± 0.66
	R	3.80 c ± 0.11	0.56 b ± 0.01	12.50 c ± 0.09	6.40 b ± 0.24	10.94 b ± 0.29
*C. morifolium*	UV-A	4.37 d ± 0.10	0.40 c ± 0.01	8.66 c ± 0.20	13.40 c ± 0.51	3.08 c ± 0.26
	B	17.91 a ± 0.17	1.84 a ± 0.02	30.36 a ± 1.29	22.60 a ± 0.68	5.28 b ± 0.15
	G	14.08 b ± 0.23	1.39 b ± 0.24	19.20 b ± 0.13	20.60 b ± 0.75	5.94 a ± 0.18
	R	12.87 c ± 0.24	1.38 b ± 0.02	18.84 b ± 0.20	23.20 a ± 0.58	5.26 b ± 0.14

Ultraviolet (UV-A), blue (B), green (G), and red (R) light. Values are means-standard errors. Means with different letters are statistically different (*p* < 0.05; *n* = 15) as determined by Duncan’s post-hoc test.

**Table 2 plants-12-02964-t002:** Content of basic metabolites and photosynthetic pigments in *S. floribundum* and *C. morifolium* grown under UV-A, blue, green, or red single-spectrum LEDs for 60 days.

Species	Light Quality	Soluble Sugar (mg·g^−1^ FW)	Soluble Protein (mg·g^−1^ FW)	Chl a (mg·g^−1^ FW)	Chl b (mg·g^−1^ FW)	Carotenoid (mg·g^−1^ FW)	Chl a/b
*S. floribundum*	UV-A	1.00 d ± 0.02	30.94 c ± 0.34	0.98 b ± 0.06	0.29 c ± 0.02	0.24 a ± 0.01	3.37 a ± 0.33
	B	3.12 b ± 0.09	37.35 b ± 1.32	1.06 b ± 0.05	0.29 c ± 0.01	0.23 a ± 0.01	3.69 a ± 0.11
	G	1.69 c ± 0.03	33.26 c ± 0.68	1.18 ab ± 0.03	0.34 b ± 0.01	0.23 a ± 0.01	3.49 a ± 0.16
	R	4.06 a ± 0.11	43.38 a ± 1.64	1.30 a ± 0.10	0.42 a ± 0.01	0.23 a ± 0.01	3.11 a ± 0.21
*C. morifolium*	UV-A	0.67 b ± 0.05	10.15 d ± 0.54	0.39 c ± 0.04	0.16 d ± 0.01	0.10 c ± 0.01	2.41 b ± 0.33
	B	0.55 bc ± 0.01	25.81 a ± 0.51	0.92 a ± 0.05	0.40 a ± 0.01	0.11 bc ± 0.01	2.28 b ± 0.08
	G	0.87 a ± 0.06	14.90 c ± 0.38	0.70 b ± 0.01	0.25 b ± 0.01	0.13 b ± 0.01	2.81 b ± 0.11
	R	0.49 c ± 0.02	19.00 b ± 0.30	0.88 a ± 0.02	0.21 c ± 0.01	0.20 a ± 0.01	4.23 a ± 0.12

Ultraviolet (UV-A), blue (B), green (G), and red (R) light. Fresh weight (FW), chlorophyll (Chl). Values are means-standard errors. Means with different letters are statistically different (*p* < 0.05; *n* = 3) as determined by Duncan’s post-hoc test.

**Table 3 plants-12-02964-t003:** Quantum yields and efficiencies/probabilities in the leaves of *S. floribundum* and *C. morifolium* grown under UV-A, blue, green, or red LEDs for 60 days.

Species	Light Quality	φ_Po_	ψ_o_	φ_Eo_	φ_Ro_	φ_Do_
*S. floribundum*	UV-A	0.782 a ± 0.004	0.630 b ± 0.023	0.493 b ± 0.020	0.147 b ± 0.020	0.219 b ± 0.004
	B	0.774 a ± 0.007	0.802 a ± 0.026	0.621 a ± 0.024	0.263 a ± 0.037	0.226 b ± 0.007
	G	0.799 a ± 0.002	0.787 a ± 0.004	0.628 a ± 0.004	0.288 a ± 0.014	0.201 b ± 0.002
	R	0.681 b ± 0.036	0.623 b ± 0.032	0.426 b ± 0.042	0.158 b ± 0.028	0.319 a ± 0.036
*C. morifolium*	UV-A	0.792 ab ± 0.005	0.710 b ± 0.021	0.562 b ± 0.015	0.163 b ± 0.007	0.208 ab ± 0.005
	B	0.803 a ± 0.001	0.782 a ± 0.014	0.628 a ± 0.012	0.245 a ± 0.020	0.197 b ± 0.001
	G	0.790 b ± 0.002	0.761 a ± 0.011	0.602 a ± 0.010	0.208 ab ± 0.012	0.210 a ± 0.002
	R	0.794 ab ± 0.004	0.778 a ± 0.003	0.618 a ± 0.002	0.249 a ± 0.022	0.206 ab ± 0.004

Ultraviolet (UV-A), blue (B), green (G), and red (R) light. Values are means-standard errors. Means with different letters are statistically different (*p* < 0.05; *n* = 5) as determined by Duncan’s post-hoc test.

**Table 4 plants-12-02964-t004:** Specific energy fluxes per reaction centers (RC) and performance indexes in *S. floribundum* and *C. morifolium* grown under UV-A, blue, green, or red LEDs for 60 days.

Species	Light Quality	ABS/RC	DI_o_/RC	TR_o_/RC	ET_o_/RC	RE_o_/RC	PI_abs_	PI_total_
*S. floribundum*	UV-A	1.391 ab ± 0.084	0.305 b ± 0.024	1.087 ab ± 0.060	0.201 b ± 0.014	0.682 b ± 0.015	4.569 b ± 0.449	2.044 b ± 0.193
	B	1.220 b ± 0.018	0.276 b ± 0.012	0.943 bc ± 0.006	0.319 a ± 0.042	0.757 a ± 0.022	12.017 a ± 1.205	9.457 a ± 0.659
	G	1.129 b ± 0.034	0.227 b ± 0.007	0.901 c ± 0.027	0.325 a ± 0.016	0.709 ab ± 0.025	12.995 a ± 0.176	11.091 a ± 0.976
	R	1.673 a ± 0.197	0.548 a ± 0.044	1.125 a ± 0.072	0.253 ab ± 0.019	0.697 ab ± 0.011	2.507 b ± 0.300	1.596 b ± 0.215
*C. morifolium*	UV-A	1.993 ab ± 0.122	0.415 a ± 0.027	1.578 ab ± 0.098	1.116 ab ± 0.041	0.323 b ± 0.015	4.826 b ± 0.299	1.972 c ± 0.345
	B	1.693 b ± 0.101	0.335 b ± 0.022	1.359 b ± 0.079	1.061 b ± 0.046	0.414 ab ± 0.030	8.829 a ± 0.621	5.810 a ± 0.262
	G	2.079 a ± 0.097	0.436 a ± 0.020	1.643 a ± 0.078	1.251 a ± 0.064	0.433 a ± 0.040	5.845 b ± 0.478	3.057 b ± 0.092
	R	1.826 ab ± 0.069	0.376 ab ± 0.017	1.450 ab ± 0.054	1.128 ab ± 0.039	0.453 a ± 0.031	7.433 a ± 0.409	5.145 a ± 0.312

Ultraviolet (UV-A), blue (B), green (G), and red (R) light. Values are means-standard errors. Means with different letters are statistically different (*p* < 0.05; *n* = 5) as determined by Duncan’s post-hoc test.

## Data Availability

Data supporting the reported results will be available and provided upon request.

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
