# Peer review of "Effect of Different Monochromatic LEDs on the Environmental Adaptability of Spathiphyllum floribundum and Chrysanthemum morifolium"

_plants, 2023, doi:10.3390/plants12162964_

Round 1

Reviewer 1 Report

Article review Yinglong Song, Weichao Liu, Zheng Wang, Songlin He, Wenqing Jia, Yuxiao Shen, Yuke Sun, Yufeng Xu, Hongwei Wang and Wenqian Shang: Different monochromatic LEDs affect the environmental adaptability of Spathiphyllum floribundum and Chrysanthemum morifolium by regulating antioxidant enzyme activity andphotosystem.

The authors investigated the effect of various monochromatic light from LED irradiators: UV-A (397.6 nm), blue (460.6 nm), green (520.7 nm) and red (661.9 nm) light on growth, antioxidant system, photosynthetic characteristics of S. floribundum ‘Tian Jiao’ (shade-loving species) and C. morifolium ‘Huang Xiu Qiu’ (light-loving species). It was shown that the activity of SOD and CAT was higher in S. floribundum under a red LED, while the activity of SOD and CAT in C. morifolium was the highest under UV light. The efficiency of the photosynthetic electron transfer chain and photosystem activity were higher when using the green and blue irradiation spectrum in S. floribundum and the blue and red spectrum in C. morifolium. 

The setting of the experiment is not entirely clear. So, the plants were placed in a light incubator for one week (the environmental conditions in the initial  light incubator were maintained with a 12-h photoperiod, a photo synthetic photon flux density of 50 µmol•m−2•s−1, and a temperature of 23 ± 2 °C), and then transferred to 1 of 4 different light environments. It is necessary to give the light intensity levels at which the plants were subsequently grown. Speaking about the effect of UV-A on plant indicators, it should be noted that this LED source captures the blue region of the spectrum.

The results obtained indicate a different reaction of the photosynthetic apparatus and the antioxidant system of shade-tolerant and light-loving plants in response to the use of sources with different spectral composition of irradiation and the inclusion of various mechanisms of adaptation to light environmental conditions.

I think that after completion the article can be published in the journal.

Reviewer 2 Report

The article titled "Different monochromatic LEDs affect the environmental adaptability of Spathiphyllum floribundum and Chrysanthemum morifolium by regulating antioxidant enzyme activity and photosystem" by Song and co-authors presents a study that examines the changes in the antioxidant enzymes activity and photosystem during testing of four wavelengths, i.e. UV-A, Green, Blue, and Red. The results of the study demonstrate variations in the antioxidant enzyme activity and chlorophyll fluorescence. The study's objectives are clearly defined, and the overall structure is well-organized. I have a few suggestions for improvement:

 Firstly, the authors present many results, but essential and informative for the presented study will be a correlation of individual results with other parameters. Please add a new subsection in "Results" and discuss this in the next chapter, i.e. "Discussion". 

Secondly, the authors used the same words as a keyword and title. I suggest using current names used in the main text, like UV-A, Green light, Catalase or CAT, etc. 

Lastly, Please expand the abbreviations: SOD- line 165; POD - line175; CAT - line 181; NBT - line 165 or opposite, e.g. unit - line 172, because in the next part authors using "U" - line 179,; PSII in line 71 and use short name "PSII" in the 76 line. Please check carefully all text, because sometimes authors write "Light emitting diodes" (line 17) and next "Light-emitting diodes" (line 44). mL (line 152) or ml (e.g.line 147, 166).

line 120 - 661.9 nm

line 125 - a reference to Hoagland solution 

Reviewer 3 Report

This is an interesting study, and the idea of this research article “Different monochromatic LEDs affect the environmental adaptability of Spathiphyllum floribundum and Chrysanthemum morifolium by regulating antioxidant enzyme activity and photosystem”. is scientifically sound. This kind of study is expected to be helpful in the determination and designing and selection of suitable outdoor and indoor plants.

This study has some flaws, especially in the introduction, and materials and methods portion. This portion must be more clear and more precise as there is a lot of literature cited that is not related to this section. The introduction section needs some literature about the selected plants and their characteristics. Conclude the introduction section properly along with the proper problem statement and objectives. Both in the abstract and conclusion, exact information (outcome) is missing. There are some minor issues related to writing, grammar, units, and storytelling in the whole manuscript that need to recheck as well.

Some other comments that need to incorporate:

The title may be shortened, precise and specified according to the study.

L 20: Use the full form at first use, then the abbreviation and follow it for each separate section/chapter (Abstract, Introduction etc).

L 20: Use "and" after the word antioxidant system.

L 21-23: Maybe provide exact values of results (Number or percentage).

L 27-30: This sentence needs to be revised by adding a concluding statement about the behavior of both selected plants and their properties under sunny and shady conditions.

L 37-40: unclear, wordy sentence, needs to be rewritten.

L 44-46: Reference or evidence needed to support this pretty strong statement.

L 66-68: “These and countless other studies” Instead of using this generalized term, maybe provide exact sources along with references.

L 69-65: Need to connect this paragraph with previous literature. 

L 98: Need a brief description of the selected plant species and the reason for choosing these plants for the current study.

L 98- Need strong statements along with proper sources for choosing this current work.

L 100-101: Use full scientific name along with authority and English name at first use.

L 102-104: The problem statement is not clear.

Materials and Methods

Research design is missing.

111-113: How many pots were sown for each treatment?

111-113 How many plants were grown in each pot?

L 143: How chlorophyll a and b values were determined? May provide formulas as well.

L 145: which device was used for determining sugar and protein contents?

L 163: Recheck the value again.

L 198-200: Comple sentence, need to be revised.

L 192: From how many plants/leaves data were taken?

Table 1: Number of leaves (per pot or plant?).

For green, red, and blue, use the same letter start with a small letter (if not at the start of the sentence) and follow it in the whole manuscript. Also, for Ultraviolet, somewhere using UV somewhere ultraviolet, try to be consistent throughout the manuscript.

L 254: “significantly higher content” provides data as well.

Table 2: What are FW and Chl? Mention full form in the table footnote.

Figure 3, 4: Mention the full form of all abbreviations used in the figure as a footnote.

The discussion section needs to be improved by adding to the point information. 

L 511: Try to conclude briefly on the basis of observed parameters.

Growth parameters are totally missing. Also, be specific both for sunny and shady plants separately.

This is an interesting study, and the idea of this research article “Different monochromatic LEDs affect the environmental adaptability of Spathiphyllum floribundum and Chrysanthemum morifolium by regulating antioxidant enzyme activity and photosystem”. is scientifically sound. This kind of study is expected to be helpful in the determination and designing and selection of suitable outdoor and indoor plants.

This study has some flaws, especially in the introduction, and materials and methods portion. This portion must be more clear and more precise as there is a lot of literature cited that is not related to this section. The introduction section needs some literature about the selected plants and their characteristics. Conclude the introduction section properly along with the proper problem statement and objectives. Both in the abstract and conclusion, exact information (outcome) is missing. There are some minor issues related to writing, grammar, units, and storytelling in the whole manuscript that need to recheck as well.

Some other comments that need to incorporate:

The title may be shortened, precise and specified according to the study.

L 20: Use the full form at first use, then the abbreviation and follow it for each separate section/chapter (Abstract, Introduction etc).

L 20: Use "and" after the word antioxidant system.

L 21-23: Maybe provide exact values of results (Number or percentage).

L 27-30: This sentence needs to be revised by adding a concluding statement about the behavior of both selected plants and their properties under sunny and shady conditions.

L 37-40: unclear, wordy sentence, needs to be rewritten.

L 44-46: Reference or evidence needed to support this pretty strong statement.

L 66-68: “These and countless other studies” Instead of using this generalized term, maybe provide exact sources along with references.

L 69-65: Need to connect this paragraph with previous literature. 

L 98: Need a brief description of the selected plant species and the reason for choosing these plants for the current study.

L 98- Need strong statements along with proper sources for choosing this current work.

L 100-101: Use full scientific name along with authority and English name at first use.

L 102-104: The problem statement is not clear.

Materials and Methods

Research design is missing.

111-113: How many pots were sown for each treatment?

111-113 How many plants were grown in each pot?

L 143: How chlorophyll a and b values were determined? May provide formulas as well.

L 145: which device was used for determining sugar and protein contents?

L 163: Recheck the value again.

L 198-200: Comple sentence, need to be revised.

L 192: From how many plants/leaves data were taken?

Table 1: Number of leaves (per pot or plant?).

For green, red, and blue, use the same letter start with a small letter (if not at the start of the sentence) and follow it in the whole manuscript. Also, for Ultraviolet, somewhere using UV somewhere ultraviolet, try to be consistent throughout the manuscript.

L 254: “significantly higher content” provides data as well.

Table 2: What are FW and Chl? Mention full form in the table footnote.

Figure 3, 4: Mention the full form of all abbreviations used in the figure as a footnote.

The discussion section needs to be improved by adding to the point information. 

L 511: Try to conclude briefly on the basis of observed parameters.

Growth parameters are totally missing. Also, be specific both for sunny and shady plants separately.
